# Waardenburg Syndrome Type 4 in Mongolian Children: Genetic and Clinical Characterization

**DOI:** 10.3390/ijms26136258

**Published:** 2025-06-28

**Authors:** Bayasgalan Gombojav, Jargalkhuu Erdenechuluun, Tserendulam Batsaikhan, Narandalai Danshiitsoodol, Zaya Makhbal, Maralgoo Jargalmaa, Tuvshinbayar Jargalkhuu, Yue-Sheng Lu, Pei-Hsuan Lin, Jacob Shu-Jui Hsu, Cheng-Yu Tsai, Chen-Chi Wu

**Affiliations:** 1Department of Epidemiology and Biostatistics, School of Public Health, Mongolian National University of Medical Sciences, Ulaanbaatar 14210, Mongolia; bayasgalan.g@mnums.edu.mn; 2International Cyber Education Center, Graduate School, Mongolian National University of Medical Sciences, Ulaanbaatar 14210, Mongolia; 3Department of Otolaryngology, School of Medicine, Mongolian National University of Medical Sciences, Ulaanbaatar 14170, Mongolia; jargalkhuu@mnums.edu.mn; 4The EMJJ Otolaryngology Hospital, Ulaanbaatar 14210, Mongolia; tsek2036@gmail.com (T.B.); zaya782000@yahoo.com (Z.M.); jmaralgoo0313@gmail.com (M.J.); j.tuvshinbayar.t@gmail.com (T.J.); 5Department of Probiotic Sciences for Preventive Medicine, Graduate School of Biomedical and Health Sciences, Hiroshima University, Hiroshima 7348551, Japan; naraa@hiroshima-u.ac.jp; 6Department of Otolaryngology, National Taiwan University Hospital, Taipei 100225, Taiwan; g05092lu@gmail.com (Y.-S.L.); peihsuanlin@ntu.edu.tw (P.-H.L.); 7Graduate Institute of Medical Genomics and Proteomics, National Taiwan University College of Medicine, Taipei 100233, Taiwan; jacobhsu@ntu.edu.tw; 8Department of Otolaryngology, National Taiwan University College of Medicine, Taipei 100225, Taiwan; 9Department of Medical Research, National Taiwan University Hospital Hsin-Chu Branch, Hsinchu 302058, Taiwan

**Keywords:** Waardenburg syndrome, *SOX10* variant, Mongolian population, de novo variant, sensorineural hearing loss, Hirschsprung’s disease

## Abstract

Waardenburg syndrome (WS) is a rare genetic disorder that affects both hearing and pigmentation. The wide divergence of WS poses significant diagnostic and management challenges. This study investigated type 4 WS within an underrepresented Mongolian population. Whole-exome sequencing revealed that two unique heterozygous variants were identified in the *SOX10* gene: c.393C>G (p.Asn131Lys) in a five-year-old female patient presenting with profound sensorineural hearing loss (SNHL), dystopia canthorum, and a white forelock; and c.535A>T (p.Lys179Ter) in a nine-year-old male patient presenting with profound SNHL, dystopia canthorum, and Hirschsprung’s disease. Temporal bone imaging revealed abnormalities in the inner ear structure in both patients. The genotypic and phenotypic characteristics were meticulously delineated, incorporating the deleterious effects of these variants, as evaluated by multiple predictive tools and the American College of Medical Genetics and Genomics (ACMG) criteria. In addition, structural characterizations were also presented using AlphaFold. The findings of this study contribute valuable genetic data to the limited literature on type 4 WS within this ethnic group and highlight the importance of genetic testing and multidisciplinary care for this rare disorder in settings with limited resources.

## 1. Introduction

Waardenburg syndrome (WS) is a rare genetic disorder that primarily affects hearing and pigmentation. The condition arises from the abnormal migration of neural crest cells during embryonic development, leading to diverse clinical manifestations, including craniofacial anomalies, pigmentation abnormalities, and sensorineural hearing loss (SNHL). The estimated prevalence of WS is approximately 1 in 42,000 individuals [1], and it can be inherited in both autosomal dominant and autosomal recessive patterns, exhibiting variable penetrance rates [2,3,4,5].

The diagnosis of WS is dependent upon a set of diagnostic criteria established by the Waardenburg Consortium, which incorporates both major and minor criteria. The major criteria encompass abnormal iris pigmentation (e.g., heterochromia), congenital SNHL, a white forelock, and dystopia canthorum (increased distance between the inner canthi), as measured by the W index [6,7]. Minor criteria include congenital leukoderma, synophrys (confluent eyebrows), a broad nasal root, hypoplastic alae nasi, and premature hair graying [6,7].

WS is classified into four distinct subtypes, each of which exhibits unique clinical features. The clinical feature of type 1 WS is characterized by dystopia canthorum, heterochromia iridis, SNHL, and pigmentary abnormalities. The diagnosis of this condition necessitates the presence of two major criteria (or one major and two minor criteria) [8]. Type 2 WS shares numerous features with type 1 but specifically lacks dystopia canthorum and requires two major criteria, including premature graying, for diagnosis [9,10,11]. Type 3 WS, otherwise referred to as Klein–Waardenburg syndrome, is a condition that exhibits the features of type 1 in conjunction with musculoskeletal deformities [12]. Type 4 WS, otherwise known as Waardenburg–Shah syndrome, is characterized by three distinct features: pigmentary abnormalities, SNHL, and Hirschsprung’s disease, a neurodevelopmental disorder leading to gastrointestinal complications due to the absence of ganglion cells in the distal colon, resulting in functional intestinal obstruction [13,14,15]. Type 1 WS and type 2 WS are the most common, accounting for approximately half and one-third of cases, respectively. Type 3 WS and type 4 WS are rare, with type 4 accounting for about 20% of cases and type 3 being exceedingly uncommon, accounting for less than 2% [3,16,17].

The diagnosis of WS is challenging due to the rarity and phenotypic variability of the condition. The identification of the causative genes and their variants is imperative for the accurate diagnosis and genetic counseling of patients. This study delineates two cases of type 4 WS in Mongolian children, both of which exhibited *SOX10* variants, as identified through whole-exome sequencing (WES). The objectives of this study are threefold: firstly, to describe the clinical presentations of the disorder; secondly, to explore the underlying pathophysiology; and thirdly, to discuss the diagnostic and therapeutic challenges associated with managing this rare disorder, especially in resource-limited settings.

## 2. Results

### 2.1. Clinical and Genetic Findings

Genetic diagnosis was achieved in two Mongolians with type 4 WS, both of whom had pathogenic variants in the *SOX10* gene.


**Case 1**


A 5-year-old female patient was referred for the evaluation of atypical blue eyes (Figure 1a) and suspected hearing impairment. She is the third child in her family; her parents and siblings have normal hearing and typical eye color. The patient exhibited profound SNHL (Figure 1c) and a substantially delayed development of language skills. The patient’s medical history was unremarkable, with the exception of constipation. There was no record of significant childhood illnesses or maternal infections during pregnancy. Physical examination further revealed the presence of dystopia canthorum (Figure 1a), white forelock (Figure 1b), and patches of hypopigmented skin. Temporal bone high-resolution computed tomography (HRCT) revealed a flattened cochlea and semicircular canal dysplasia (Figure 1d).

A de novo heterozygous *SOX10* variant (c.393C>G, p.Asn131Lys) was identified in Case 1 through WES assays but was not detected in her parents or other family members (Figure 1e), consistent with an autosomal dominant pattern, as reported in previous studies [18,19,20] and curated in the database (OMIM #602229). As demonstrated in Table 1, the *SOX10* c.393C>G variant was not found in population genome databases; however, it was catalogued as “pathogenic” in the ClinVar database (https://www.ncbi.nlm.nih.gov/clinvar/ (accessed on 31 May 2025)). The deleterious effect of pathogenicity was assessed via multiple predictive tools, in conjunction with our assertions based on the American College of Medical Genetics and Genomics (ACMG) guideline. The residue Asn131 of the *SOX10*-encoded protein was demonstrated to be a highly evolutionarily conserved site across multiple species (eight common animal models taken as examples in Figure 2). The PhyloP100way score (data last updated on 8 May 2015 curated in UCSC [University of California Santa Cruz] Genome Browser) of this genomic locus (*chr22:37983392* of hg38 for *SOX10* c.393C>G) exhibited mild conservation (2.14, ranging from −1.448 to 9.768). This result can be attributed to the prevalence of the two predominant ATT and GTT (underlined bases indicate the targeted variant on the negative strand) at this site across species, which form silent polymorphisms to encode the same amino acid Asn. This result indicates the high degree of conservation of residue 131.

At the age of five, the patient underwent a cochlear implantation procedure in her right ear. This delayed intervention was attributed to the absence of newborn hearing screening, resulting in a later identification of her hearing loss, primarily due to delayed speech acquisition and financial constraints, as the procedure was not initially covered by the patient’s insurance. Preoperative auditory brainstem response (ABR) measurements revealed bilateral profound sensorineural hearing loss. A Cochlear firm CI 422 electrode was used for the implantation. Intraoperative measurements showed fair impedance and robust electrically evoked compound action potentials (ECAPs) across all 22 electrodes. Postoperative measurements confirmed fair impedance and robust ECAPs across all 22 electrodes. Postoperative audiological results showed aided hearing thresholds between 20 and 30 dB, a speech recognition threshold of 30 dB, and a word recognition score of 75% at 60 dB. In consideration of the patient’s age and clinical stability, no immediate medical intervention beyond cochlear implantation was scheduled. Genetic counseling was recommended for the patient and her family to discuss the implications of WS, potential disease progression, and management strategies. In order to ensure the ongoing monitoring of the patient’s hearing function, vision, and the potential complications associated with WS, regular follow-up appointments with an otolaryngologist, audiologist, and ophthalmologist were scheduled.


**Case 2**


A nine-year-old male patient presented with dystopia canthorum (Figure 3a) but had brown eyes, which was consistent with the eye color of his family. The subject’s mother and sister demonstrated normal hearing, whereas his father was deaf and had undergone a cochlear implant procedure. His medical history was notable only for constipation, with no other significant medical issues. Born in Korea, his hearing impairment was identified shortly after birth. Audiological evaluations confirmed bilateral profound SNHL at six months of age (Figure 3b). Upon reevaluation of the ABR results, the fifth wave was absent at 100 dB on click ABR in the right ear. On the left side, the fifth wave was present but unstable at 100 dB, with a latency of 7.4 ms, indicating an insecure response. All family members, with the exception of his father, demonstrated normal hearing, and no other significant health problems were reported in the family. The diagnosis of Waardenburg–Shah syndrome (type 4 WS) was made on the basis of a paternal family history of deafness, biopsy-proven Hirschsprung’s disease, and bilateral SNHL. HRCT imaging of the temporal bone revealed a flattened cochlea and semicircular canal dysplasia (Figure 3c).

Genetic testing identified a heterozygous *SOX10* variant (c.535A>T, p.Lys179Ter) (Figure 3d) in Case 2. This variant was regarded as the causative variant following an autosomal dominant pattern. As demonstrated in Table 1, *SOX10* c.535A>T was also not found in the population genome and disease-related variation databases. This premature termination variant is predicted to cause a truncated translation, resulting in severe pathogenicity, based on our ACMG-guided assertions. The Lys179 residue also exhibited high evolutionary conservation, with a PhyloP100way score of 7.96 (ranging from −7.306 to 9.867) at a locus of *chr22: 37978029* in hg38 for *SOX10* c.535A>T.

The patient underwent surgery for Hirschsprung’s disease, which involved a laparoscopic endorectal pull-through procedure, followed by the administration of postoperative broad-spectrum antibiotics. Bilateral cochlear implantation was conducted at two years of age using the Cochlear firm CI 422 electrode. Intraoperative measurements revealed fair impedance and robust ECAPs across all 22 electrodes in both ears. Postoperative measurements confirmed fair impedance and robust ECAPs across all 22 electrodes in both ears. Postoperative audiological results showed aided hearing thresholds between 20 and 30 dB in both ears, a speech recognition threshold of approximately 30 dB, and a word recognition score of 75% at 60 dB. In order to ensure the optimal monitoring of the patient’s auditory and visual capacities and potential complications of Waardenburg–Shah syndrome, a structured program of regular follow-up consultations with the respective specialties of otolaryngology, audiology, and ophthalmology was scheduled.

### 2.2. Structural and Pathogenic Analysis of SOX10 Variants

In Case 1, the causative variant *SOX10*: c.393C>G (p.Asn131Lys) is located within the high-mobility group (HMG) domain, which has been previously reported to be enriched with hotspots [22]. This local region comprises a highly conserved DNA-binding domain that is essential for the function of transcription factors in various cellular developmental processes [23,24]. The functionality of this domain was also accurately predicted in the UniProt (Figure 4a) and AlphaFold (Figure 4b) databases. Further investigation of the predicted human *SOX10* conformation curated in the AlphaFold Protein Database (ID: AF-P56693-F1-v4) revealed that Asn131 resides at the interface of the helix–loop–helix domain within a high-confidence molecular coordinate region (Figure 4c). The entire red-highlighted domain represents a highly pathogenic region, as determined by the average predicted scores in the AlphaMissense saturation mutagenesis table (Figure 4d). The variant p.Asn131Lys in this table also exhibited a high pathogenicity score. The observed pathology may arise from the substitution destabilizing surrounding residues and leading to the increased repulsion of positively charged residues (e.g., Arg119) and hydrophobic residues (e.g., Leu134). Further exploration of conformational changes induced by this point mutation will be necessary once a high-resolution *SOX10* conformation is determined.

In Case 2, the causative variant *SOX10*: c.535A>T (p.Lys179Ter) is predicted to result in premature termination. This results in the disruption of the essential downstream transcriptional domain (residues 228–310; see Figure 4a) and consequent inactivation of the normal function of the *SOX10*-encoded transcription factor. Subsequent functional assays should be conducted in order to confirm the loss-of-function effect of this variant.

## 3. Discussion

WS encompasses a group of rare congenital genetic disorders, which are primarily inherited in an autosomal dominant pattern. The estimated prevalence of WS is approximately 1 in 42,000 [1], accounting for over 2% of congenital deafness cases [16]. The pathogenesis involves abnormal melanocyte distribution during embryonic development and genetic variants implicated in neural crest cell function. This results in a wide range of clinical presentations, leading to the classification into four distinct subtypes. Type 4 WS is a relatively rare subtype, with an estimated prevalence of 1 in 1,000,000 [25]. Documentation of type 4 WS is limited, with only about 50 cases having been recorded to date, thus underscoring its rarity [26].

The diagnosis of Waardenburg syndrome can be challenging due to its phenotypic variability and the paucity of published data, especially with regards to the rarer subtypes. An accurate diagnosis and early intervention, with the potential to mitigate morbidity significantly, necessitate a comprehensive family history, clinical examination, and audiological evaluation of first-degree relatives [27,28,29]. Research has identified the presence of pathogenic variants in multiple subtypes of WS. Types 1 and 3 WS are frequently associated with variants in the *PAX3* gene [30,31,32]. Type 2 WS has been linked to pathogenic variants in the *MITF*, *SNAI2*, and *SOX10* genes [33,34]. Type 4 WS has been found to be predominantly associated with pathogenic variants in the *EDN3*, *EDNRB*, and *SOX10* genes [35,36,37,38] and, less commonly, with bi-allelic variants in the *MITF* gene [39].

The genetic heterogeneity of WS4 is characterized by the presence of three distinct etiological subtypes: WS4A, WS4B, and WS4C. Research indicates that variants in the *EDNRB* gene, located on chromosome 13q22.3 and encoding the endothelin-B receptor, are associated with a 53.3% prevalence of SNHL in patients with WS4A. For instance, a patient from a family without a history of Hirschsprung’s disease exhibited novel homozygous missense variants at codon 196 in exon 2 (p.Ser196Asn), whereas both parents and four of six siblings presented with a heterozygous variant but lacked clinical manifestations [40]. Syrris et al. identified a novel nonsense variant at codon 253 (CGA → TGA, Arg → STOP) in *EDNRB*, leading to the premature termination of translation at exon 3 [41]. Consequently, *EDNRB* variants demonstrate a more intricate interaction than that exhibited by simple dominant or recessive traits, thereby suggesting that homozygous variants may result in isolated Hirschsprung’s disease, while heterozygous variants may be associated with WS4. WS4B is associated with variants in the *EDN3* gene, located on chromosome 20q13.32. The *EDN3* gene encodes a ligand for the endothelin receptor, which is crucial for neural crest cell development, including enteric nervous system and melanocyte formation. WS4C has been linked to variants in the *SOX10* gene, located on chromosome 22q13. This gene plays a pivotal role in melanocyte development, regulating *MITF* expression in collaboration with *PAX3* [42]. In human subjects, *SOX10* variants have been documented as the underlying cause of approximately 15% of WS2 cases and 40–50% of WS4 patients [20]. As demonstrated by Verheij et al. [43] and Fernández et al. [44], *SOX10* variants have been shown to be associated with a more severe phenotype, termed PCWH (peripheral demyelinating neuropathy, central demyelinating leukodystrophy, Waardenburg syndrome, and Hirschsprung’s disease), further supporting the genetic and phenotypic heterogeneity. Beyond direct genetic variations, insights from mouse models suggest that complex genetic mechanisms, such as regulatory element disruption, can also contribute to WS4-like phenotypes [45].

A systemic review of *SOX10* variants, as conducted by Pingault et al. [22], revealed the presence of over 25 premature termination variants, which were found to be distributed throughout the gene, and 14 missense variants, which were predominantly enriched in the high-mobility group (HMG) domain. Less frequent variants included sporadic mis-splicing or start codon alterations. Premature termination variants accounted for the majority of cases across *SOX10*-related disorders. In this study, we detailed two additional heterozygous *SOX10* variants identified in two unrelated patients and elucidated their genotypic and phenotypic characteristics. Case 1’s causative variant, *SOX10*: c.393C>G (p.Asn131Lys), is located in the highly conserved HMG domain that adopts a helix–loop–helix conformation (AlphaFold ID: AF-P56693-F1-v4). This variant exhibits a high AlphaMissense score, indicating the presence of strong pathogenic properties in the event of an abnormal substitution. Conversely, the causative variant *SOX10*: c.535A>T (p.Lys179Ter), detected in Case 2, is predicted to result in premature termination. This truncation is predicted to result in a severe impairment of the essential downstream transcriptional domain (residues 228–310), consequently leading to the inactivation of the normal function of the *SOX10*-encoded transcription factor. It is imperative that future experimental validation is undertaken to confirm the pathological mechanisms and loss-of-function effects of both variants.

Notably, our patients demonstrated good CI performance, despite their malformed cochleae. This finding aligns with previous research indicating favorable CI outcomes in patients with Waardenburg syndrome, regardless of the specific genetic variation [46]. One plausible explanation for this success is that, even with a malformed cochlea, the spiral ganglion neurons—which are crucial for CI function—remain largely intact in *SOX10* patients [47,48]. Furthermore, a regular, slim electrode such as the Cochlear firm CI 422, which was used in our cases, appears to be sufficient, even in the presence of a flattened cochlea. Therefore, cochlear implantation is a viable rehabilitation option for patients with *SOX10* variations.

The multifaceted clinical manifestations associated with WS necessitate a comprehensive multidisciplinary management strategy. Cochlear implantation and associated support services are of paramount importance in the rehabilitation of SNHL patients. Hirschsprung’s disease necessitates surgical intervention, most commonly achieved by means of an endorectal pull-through procedure, to address the underlying cause of intestinal obstruction. This case series provides valuable insights into WS4 within an underrepresented Mongolian population. A significant strength of this study is the identification of two heterozygous *SOX10* variants in these patients, which contributes to the expanding knowledge of the disorder’s genetic basis. The emphasis on the Mongolian population is particularly noteworthy, as it addresses the paucity in genetic data across diverse ethnic groups. Nevertheless, some limitations of this study merit discussion. The small sample size in this study, with only two cases in total, restricts the generalizability of the findings. Furthermore, while genetic variants were identified, the study lacks functional analyses to elucidate the precise impact of these variants on protein function and disease pathogenesis. Nonetheless, this report augments the sparse literature on type 4 WS and highlights the critical role of genetic testing and multidisciplinary care, even within resource-constrained settings.

## 4. Materials and Methods

### 4.1. Subjects and Clinical Assessments

Patients diagnosed with type 4 WS in Mongolia were recruited for this study. Comprehensive clinical data were compiled, encompassing patient histories, physical examinations, and audiological evaluations. Pedigree charts were constructed to present family history and inheritance patterns. The clinical features that were recorded included eye color, the presence of a white forelock, dystopia canthorum, and skin pigmentation. The presence and severity of SNHL were determined through audiometric evaluations relevant to patients’ age. Informed consent was obtained from all participants and/or their legal guardians. Auditory brainstem response (ABR) testing was performed using the Interacoustics Eclipse system. Click stimuli were presented at 90 and 100 dB nHL with alternating polarity. Responses were filtered between 100 and 3000 Hz and averaged over 1500–2000 sweeps, with a 20 ms analysis window. The absence of definable Wave V at 100 dB indicated profound sensorineural hearing loss. Non-auditory artifacts, such as post-auricular muscle responses observed at certain latencies (e.g., around 2 or 14 ms), were differentiated from true auditory neural activity by the audiologist. High-resolution computed tomography (HRCT) was performed using a SOMATOM go.NOW^®^ CT scanner (Siemens Healthineers, Forchheim, Germany) to assess the inner ear structures with a resolution of 0.6 mm. The study was granted ethical approval by the NTUH Research Ethics Committee (202007065RINB) and the EMJJ Otolaryngology Hospital of Mongolia (Medical Ethics Committee of the Ministry of Health, Mongolia, No: 23/065).

### 4.2. Genetic Examination with Multi-Database Assessment for Pathogenicity

A WES assay, as previously reported [21], was performed to identify and characterize pathogenic variants in the cases of type 4 WS. The variant pathogenicity was assessed using the Varsome platform (ver. 13.4.0, https://varsome.com/ (accessed on 30 May 2025)) [49]. The Varsome bioinformatic dataset (dbNSFP ver. 4.9, https://usf.app.box.com/s/535gi9e8twzv2wr5j2sxlto3i6tc208d (accessed on 30 May 2025)) contained several in silico predictive results, including CADD (ver. 1.7) [50], AlphaMissense (ver. 03-Jul-2024) [51], SIFT (ensembl 66, released in January 2015) [52], PolyPhen-2 (ver. v2.2.2) [53], and REVEL (released on 3 May 2021) [54]. Curated pathogenic assertions of the targeted variants were enquired from disease databases, including ClinVar (https://www.ncbi.nlm.nih.gov/clinvar/ (accessed on 31 May 2025)) [55] and the Deafness Variants Database (DVD, ver. 9; last accessed on 13 May 2025) [56]. Allele frequencies of the targeted variants were queried in the gnomAD population database (ver. 4.1) [57]. Evolutionary conservation at specific genomic loci was accessed using the PhyloP100way score [58] (data last updated on 8 May 2015 curated in the UCSC [University of California Santa Cruz] Genome Browser [59]). The American College of Medical Genetics and Genomics (ACMG) guideline [60] was applied to classify variant pathogenicity.

### 4.3. Homologous Sequence Analyses and Structural Characterizations

Homologous genes across species were explored using the BLAT search tool in the UCSC Genome Browser (https://genome.ucsc.edu/cgi-bin/hgBlat (accessed on 30 May 2025)). Multiple sequence alignment was performed using the Clustal Omega program (ver. 1.2.4) on the EMBL-EBI website (https://www.ebi.ac.uk/jdispatcher/msa/clustalo (accessed on 30 may 2025)). Structural domains of the targeted deafness gene were queried from UniProt (https://www.uniprot.org/ (accessed on 30 may 2025)) [61]. The predicted structure of the targeted deafness gene was visualized and analyzed in the AlphaFold Protein Structure Database (https://alphafold.ebi.ac.uk/ (accessed on 30 may 2025)) [62], which contains the AlphaMissense saturation mutagenesis heatmap used to visualize pathogenicity scores along the protein sequence.

## 5. Conclusions

This study presents two rare cases of WS4 in Mongolian children, confirmed through clinical evaluation, family history, and the identification of unique *SOX10* variants (c.393C>G [p.Asn131Lys] and c.535A>T [p.Lys179Ter]). Our findings underscore the significance of genetic testing in the diagnosis of this rare disorder, particularly among underrepresented populations, and contribute to the expansion of the limited data on the genetic spectrum of WS4. These cases highlight the imperative for a multidisciplinary approach and genetic counseling to optimize patient outcomes in resource-constrained settings, where accurate diagnosis and tailored management are of paramount importance.

## Figures and Tables

**Figure 1 ijms-26-06258-f001:**
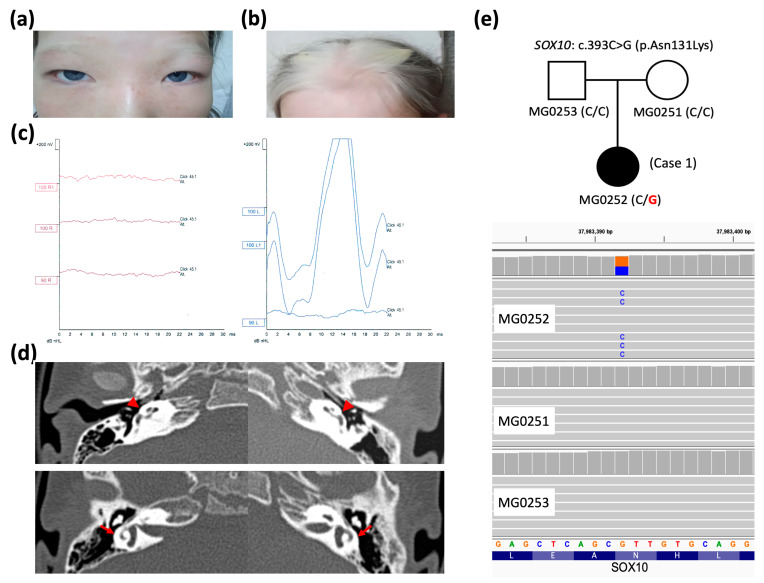
Clinical features, imaging, and genetic data of Case 1. (**a**,**b**) Dystopia canthorum, white forelock, and heterochromia iridis, as seen in the clinical photographs. (**c**) Evidence of profound sensorineural hearing loss (>100 dB) identified in auditory brainstem response examinations. (**d**) Bilateral flattened cochleae (red arrowheads) and dysplastic semicircular canals with small central bony islands (red arrows) are shown in high-resolution computed tomography of the temporal bone. (**e**) Pedigree of Case 1 (affected patient colored in black) showing a de novo *SOX10* variant c.393C>G (wild-type nucleotide in black and variation in red at – strand of hg38 genome) confirmed via whole-exome sequencing (wild-type nucleotide in orange and variation in blue at + strand of hg38 genome).

**Figure 2 ijms-26-06258-f002:**
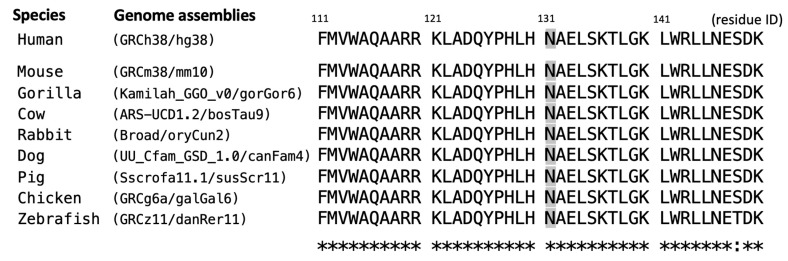
Multiple sequence alignment of the *SOX10*-encoded protein across different species in the flanking regions of Asn131 (colored in gray). Asterisks (*) denote the fully-conserved position with the same residue; colon (:) indicate the highly-conserved position with residue in similar properties.

**Figure 3 ijms-26-06258-f003:**
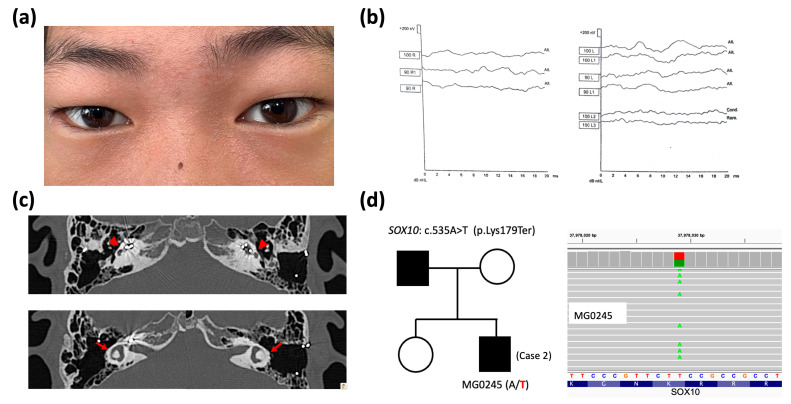
Clinical features, imaging, and genetic data of Case 2. (**a**) Dystopia canthorum observed in a clinical photograph. (**b**) Profound sensorineural hearing loss (>100 dB) in auditory brainstem response examinations. (**c**) Bilateral flattened cochleae (red arrowheads) with cochlear implant electrodes inside and dysplastic semicircular canals (red arrows) are shown in high-resolution computed tomography of the temporal bone. (**d**) Pedigree of Case 2 (affected patients in black) showing a heterozygous *SOX10* variant c.535A>T (wild-type nucleotide in black and variation in red at – strand of hg38 genome) confirmed via whole-exome sequencing (wild-type nucleotide in red and variation in green at + strand of hg38 genome).

**Figure 4 ijms-26-06258-f004:**
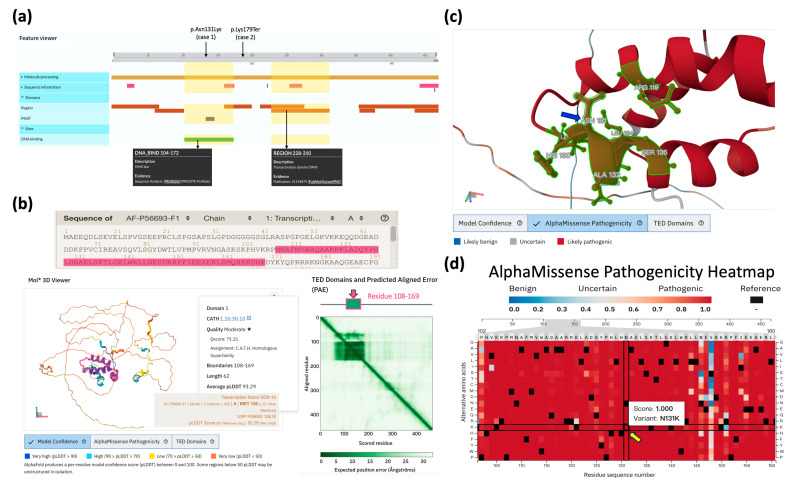
Overview of structural domains and deleterious effects of *SOX10* variants. (**a**) Predicted *SOX10* domains derived from the UniProt database (Entry ID: P56693). (**b**) The AlphaFold-based human *SOX10* conformation with predicted position error (ver. AF-P56693-F1-v4), whose high-confidence domain (residue 108–169 with highly-precious predicted coordinates [dark green] marked in protein sequence and conformation [pink]) is determined as the high-mobility group (HMG) box domain (CATH Homologous Superfamily 1.10.30.10). (**c**) An illustrative image of the local *SOX10* conformation surrounding Asn131 (blue arrow), as visualized in the AlphaFold structural viewer. (The green region is within 5 Å of residue 131.) (**d**) The AlphaMissense saturation mutagenesis heatmap curated in the AlphaFold database shows the pathogenicity scores along the human *SOX10* protein sequence, where the variant p.Asn131Lys (yellow arrow) suggests high damaging score of 1 (score between 0 and 1 that higher value indicating a greater likelihood of pathogenic effect).

**Table 1 ijms-26-06258-t001:** Pathogenic *SOX10* variants in the two cases.

Patients	Loci (hg38) and HGVS ‡ of Variants	Grpmax Allele Frequency §	Predictive Scores ¶	Database Assertions	ACMG Criteria (Classification)	Ref.
Case 1 (MG0252)	*chr22-37983392-G-C* *SOX10*: c.393C>G (p.Asn131Lys)	N.A.	CADD: 27.8 AlphaMissense: 1.0 (PS) SIFT: 0.001 (D) POL2: 0.998 (PD) REVEL: 0.917 (PS)	ClinVar: P DVD: N/A	PS1, PS2, PM2, PP3, PP4 (Pathogenic)	[21]
Case 2 (MG0245)	*chr22-37978029-T-A* *SOX10*: c.535A>T (p.Lys179Ter)	N.A.	CADD: 37 BayesDel: 0.994 (PS)	ClinVar: N/A DVD: N/A	PVS1, PM2, PP4 (Pathogenic)	[21]

**‡** The variant description is based on the Human Genome Variation Society (HGVS) nomenclature using MANE (Matched Annotation from NCBI and EMBL-EBI) select transcript (NM_006941.4) for *SOX10* variants. § Grpmax-AF: The maximum allele frequency across populations in the gnomAD database (ver. 4.1). ¶ abbreviations: N.A.: Not available; D, damaging; PD, probably damaging; PS, pathogenicity strong; ACMG: American College of Medical Genetics and Genomics.

## Data Availability

The datasets used and/or analyzed during the current study are available from the corresponding authors upon reasonable request.

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
