# Peer review of "Waardenburg Syndrome Type 4 in Mongolian Children: Genetic and Clinical Characterization"

_ijms, 2025, doi:10.3390/ijms26136258_

Round 1
Reviewer 1 Report
Comments and Suggestions for Authors
This is a very interesting article regarding very rare variaton of Waardenberg sy
Authors gave us good introduction where they explained the syndrome and the problem with establishing diagnosis.
Regarding first patient received cochlear implant at the age of 5. Authors should give us more information why is this done so late, since we now that best results are when implantation is done under age of 3 y. Since there was a malformation of cochlea what kind of electrode was used and were there any difficulties during surgery. Same explanation should be done for the second patient.
Picture 3 C – it seems that cochlear implant is inside cochlea this should be described
In discussion area authors should provide data concerning the success of cochlear implantation in previous studies since malformation of the cochlea is present.
References are relevant for this research.
Reviewer 2 Report
Comments and Suggestions for Authors
Dear authors,
thank you for submitting your manuscript about Waardenburg Syndrome Type 4.
Here are my recommandations:
Intro:
Tell the reader what the Hirschsprung desease is including infos about incidence, ... (e.g., "gastrointestinal complications due to the absence of ganglion cells in the distal colon, resulting in functional intestinal obstruction.")
Add data about the percentage of the different types of WS.
fix "specifi[8]cally"
You can implement more literature about WS type 4 using ...
Fernández, R et al. (2014). Waardenburg syndrome type 4: Report of two new cases caused by SOX10 mutations in Spain. American Journal of Medical Genetics Part A, 164, 542 - 547. https://doi.org/10.1002/ajmg.a.36302
Pang, X et al. (2018). A homozygous MITF mutation leads to familial Waardenburg syndrome type 4. American Journal of Medical Genetics Part A, 179, 243 - 248. https://doi.org/10.1002/ajmg.a.60693
Bergeron, K et al. (2016). Upregulation of the Nr2f1-A830082K12Rik gene pair in murine neural crest cells results in a complex phenotype reminiscent of Waardenburg syndrome type 4. Disease Models & Mechanisms, 9, 1283 - 1293. https://doi.org/10.1242/dmm.026773
... into your intro and discussion.
Results:
fig 1c: please provide more details about the ABR in the text (EP system, filtering, averages, stimulator, masking, ...). In addition the measurement has a wrong scaling. 2µV is far to high for results in ABR. And please discuss the result on the left ear: What would this response around 2ms be? eI? What is this response around 14ms? PAM? ...? And provide a higher quality image of this measurement report.
fig 1d: please provide info about the device used for the CT and what high-res exactly is.
case 1: please provide more details about the cochlear implantation. pre-op. audiogram, used implant, results of intra-op. measruements, post-op. measurements like impedance and post-op. results. this data is of high intereest in the field of cochlear implants.
Fig.2: Please provide a higher res.
Fig.3a: this sub-figure could focus more on the patients eyes to increase anonymity without a loss of info for the reader.
Fig.2b: These ABR waveforms are much better. sadly, there is no second measrument at 100 R. And in 100 L and 90 L, using alternating results, I'm not 100% sure that there is no response. At least an insecure response should be stated.
For CI: Please add the same infos requested in case 1.
